# Example-based Hypernetworks for Multi-source Adaptation to Unseen Domains

**Tomer Volk**[*1]**, Eyal Ben-David**[*1]**, Ohad Amosy**[2]**, Gal Chechik**[2 3]**, Roi Reichart**[1]

[1]Faculty of Data and Decision Sciences, Technion, IIT
[2]Bar Ilan University, Israel
[3]NVIDIA Research
{tomervolk|eyalbd12|roiri}@technion.ac.il
{amosyoh|gal.chechik}@biu.ac.il

## Abstract

As Natural Language Processing (NLP) algorithms continually achieve new milestones, out-of-distribution generalization remains a significant challenge. This paper addresses the issue of multi-source adaptation for unfamiliar domains: We leverage labeled data from multiple source domains to generalize to unknown target domains at training. Our innovative framework employs *example-based Hypernetwork adaptation*: a T5 encoder-decoder initially generates a unique signature from an input example, embedding it within the source domains' semantic space. This signature is subsequently utilized by a Hypernetwork to generate the task classifier's weights. In an advanced version, the signature also enriches the input example's representation. We evaluated our method across two tasks—sentiment classification and natural language inference—in 29 adaptation scenarios, where it outpaced established algorithms. We also compare our finetuned architecture to few-shot GPT-3, demonstrating its effectiveness in essential use cases. To our knowledge, this marks the first application of Hypernetworks to the adaptation for unknown domains[1].

## 1 Introduction

Deep neural networks (DNNs) have substantially improved natural language processing (NLP), reaching task performance levels that were considered beyond imagination until recently (Conneau and Lample, 2019; Brown et al., 2020). However, this unprecedented performance typically depends on the assumption that the test data is drawn from the same underlying distribution as the training data. Unfortunately, as text may stem from many origins, this assumption is often not met in practice. In such cases, the model faces an out-of-distribution (OOD) generalization scenario, which often yields significant performance degradation.

To alleviate this difficulty, several OOD generalization approaches proposed to use unlabeled data from the target distribution. For example, a prominent domain adaptation (DA, (Daumé III, 2007; Ben-David et al., 2010)) setting is unsupervised domain adaptation (UDA, (Ramponi and Plank, 2020)), where algorithms use labeled data from the source domain and unlabeled data from both the source and the target domains (Blitzer et al., 2006, 2007; Ziser and Reichart, 2017). In many real-world scenarios, however, it is impractical to expect training-time access to target domain data. This could happen, for example, when the target domain is unknown, when collecting data from the target domain is impractical or when the data from the target domain is confidential (e.g. in healthcare applications). In order to address this setting, three approaches were proposed.

The first approach follows the idea of *domain robustness*, generalizing to unknown domains through optimization methods which favor robustness over specification (Hu et al., 2018; Oren et al., 2019; Sagawa et al., 2020; Wald et al., 2021; Zhao et al., 2020). Particularly, these approaches train the model to focus on domain-invariant features and overlook properties that are associated only with some specific source domains. In contrast, the second approach implements a domain expert for each source domain, hence keeping knowledge of each domain separately. In this *mixture-of-experts (MoE)* approach (Kim et al., 2017; Guo et al., 2018; Wright and Augenstein, 2020; Mansour et al., 2008), an expert is trained for each domain separately, and the predictions of these experts are aggregated through averaging or voting.

To bridge the gap between these opposing approaches, a third intermediate approach has been recently proposed by Ben-David et al. (2022). Their PADA algorithm, standing for a Prompt-based Autoregressive Approach for Adaptation to Unseen Domains, utilizes domain-invariant and domain-

---

*Both authors equally contributed to this work.

[1]Our code and data are available at https://github.com/TomerVolk/Hyper-PADA

| |
|---|
| **Premise.** *Homes not located on one of these roads must place a mail receptacle along the route traveled.* **Hypothesis.** *Other roads are far too rural to provide mail service to.* **Domain.** *Government.*      **Label.** *Entailment.* **DRF Signature.** *travel: city, area, town, reports, modern* |
| **Fiction:** *jon, tommy, tuppence, daan, said, looked* **Slate:** *newsweek,* ***reports****, according, robert* **Telephone:** *yeah, know, well, really, think, something* **Travel:** *century,* ***city****, island,* ***modern****,* ***town****, built,* ***area*** |

Table 1: An example of Hyper-DRF and Hyper-PADA application to an MNLI example. In this setup the source training domains are *Fiction, Slate, Telephone and Travel* and the unknown target domain is *Government*. The top part presents the example and the DRF signature generated by the models. The bottom-part presents the DRF set of each source domain.

specific features to perform *example-based* adaptation. Particularly, given a test example it generates a unique prompt that maps this example to the semantic space of the source domains of the model, and then conditions the task prediction on this prompt. In PADA, a T5-based algorithm (Raffel et al., 2020), the prompt-generation and task prediction components are jointly trained on the source domains available to the model.

Despite their promising performance, none of the previous models proposed for DA in NLP explicitly learns both shared and domain-specific aspects of the data, and effectively applies them together. Particularly, robustness methods focus only on shared properties, MoE methods train a separate learner for each domain, and PADA trains a single model using the training data from all the source domains, and applies the prompting mechanism in order to exploit example-specific properties. This paper hence focuses on improving generalization to unseen domains by explicitly modeling the shared and domain-specific aspects of the input.

To facilitate effective parameter sharing between domains and examples, we propose a modeling approach based on *Hypernetworks* (HNs, Ha et al. (2017)). HNs are networks that generate the weights of another target network, that performs the learning task. The input to the HN defines the way information is shared between training examples. Mahabadi et al. (2021) previously focused on a simpler DA challenge, applying HNs to supervised DA, when a small number of labeled examples from the target are used throughout the training procedure. Nevertheless, to the best of our knowledge, we are

the first to apply HNs to DA scenarios where labeled data from the target domain, and actually also any other information about potential future test domains, are not within reach. Hence, we are the first to demonstrate that HNs generalize well to unseen domains.

We propose three models of increasing complexity. Our basic model is Hyper-DN, which explicitly models the shared and domain-specific aspects of the training domains. Particularly, it trains the HN on training data from all source domains, to generate classifier weights in a domain-specific manner. The next model, Hyper-DRF, an example-based HN, performs parameter sharing at both the domain and the example levels. Particularly, it first generates an example-based signature as in PADA, and then uses this signature as input to the HN so that it can generate example-specific classifier weights.[2] Finally, our most advanced model is Hyper-PADA which, like Hyper-DRF, performs parameter sharing at both the example and domain levels, using the above signature mechanism. Hyper-PADA, however, does that at both the task classification and the input representation levels. For a detailed description see §3.

We follow Ben-David et al. (2022) and experiment in the any-domain adaptation setup (§4,5). Concretely, we leverage labeled datasets from multiple domains, excluding one for testing in our leave-one-out experiments. Although designed for cross-domain (CD) generalization, we can explore our models in cross-language cross-domain (CLCD) setups using a multilingual pre-trained language model. In CLCD and CD sentiment classification (12 settings each) and CD MNLI (5 settings), Hyper-PADA outperforms an off-the-shelf SOTA model (a fine-tuned T5-based classifier, without domain adaptation) by $9.5\%$, $8.4\%$, and $14.8\%$ on average, respectively. Also, our results show that our HN-based methods surpass previous models from the three families described above. Lastly, additional comparisons show the value of individual components in our HN-based algorithms, and reinforce the need for DA methods in the era of huge LMs when comparing Hyper-PADA to GPT-3.

---

[2]DRFs stand for *Domain Related Features* and DN stands for *Domain Name*. See §A.1

## 2 Related Work

### 2.1 Unsupervised Domain Adaptation

Most recent DA research addresses UDA (Blitzer et al., 2006; Reichart and Rappoport, 2007; Glorot et al., 2011). Since the rise of DNNs, the main focus of UDA research shifted to representation learning methods (Titov, 2011; Glorot et al., 2011; Ganin and Lempitsky, 2015; Ziser and Reichart, 2017, 2018, 2019; Rotman and Reichart, 2019; Han and Eisenstein, 2019; Ben-David et al., 2020; Lekhtman et al., 2021).

Our paper considers a recent DA setup, presuming no training-time knowledge of the target domain, termed as *any-domain adaptation* (ADA) by Ben-David et al. (2022)). As discussed in §1, some papers that addressed this setup follow the domain robustness path (Arjovsky et al., 2019), while others learn a mixture of domain experts (Wright and Augenstein, 2020). Ben-David et al. (2022) presented *PADA*, an algorithm trained on data from multiple domains and adapted to test examples from unknown domains through prompting. PADA leverages *domain related features (DRFs)* to implement an example-based prompting mechanism. The DRFs provide semantic signatures for the source domains, representing the similarities among them and their unique properties.

PADA is trained in a multitask fashion on source domain examples. For each example, it is trained to either generate a DRF signature or classify the example, using the signature as a prompt. Inference involves PADA generating a DRF for its input and classifying it using this signature. The addition of an architecture component, HNs, necessitates a two-phase training process, as outlined in §3. Unlike previous DA work in NLP (and specifically PADA), we perform adaptation through HNs which are trained to generate the weights of the task classifier in a domain-based or example-based manner. This framework allows us to explicitly model domain-invariant and domain-specific aspects of the data and perform example-based adaptation.

### 2.2 Hypernetworks

Hypernetworks (HNs) (Ha et al., 2017) are networks trained to generate weights for other networks, enabling diverse, input-conditioned personalized models. =In Figure 1, we present an HN-based sentiment classification model. The model receives a review that originates from the "Movies" domain and the HN ($f$), which is conditioned on

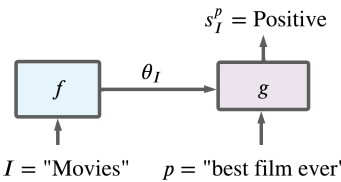

$$s_I^p = \text{Positive}$$

$I = \text{"Movies"} \qquad p = \text{"best film ever"}$

Figure 1: A discriminative model, based on hypernetworks. The HN ($f$), that is conditioned on the example domain ($I$), generates the weights ($\theta_I$) for a classifier ($g$), which is based on a feedforward network.

the domain name, generates the weights for the discriminative architecture ($g$), which, in turn, predicts the (positive) sentiment of the input review ($p$). HNs are formulated by the following equations:

$$\theta_I = f(I, \theta_f) \qquad (1)$$
$$s_I^p = g(p, \theta_I) \qquad (2)$$

Where $f$ is the HN, $g$ is the main task network, $\theta_f$ are the learned parameters of $f$, $I$ is the input of $f$, and $p$ is the representation of the input example. $\theta_I$, the parameters of network $g$, are generated by the HN $f$, and $s_I^p$ are the (task-specific) model predictions. The classification loss and it's gradients are then calculated with respect to the HN. Applied widely across computer vision (Klein et al., 2015; Riegler et al., 2015; Klocek et al., 2019), continual learning (von Oswald et al., 2020), federated learning (Shamsian et al., 2021), weight pruning (Liu et al., 2019), Bayesian neural networks (Krueger et al., 2017; Ukai et al., 2018; Pawlowski et al., 2017; Deutsch et al., 2019), multi-task learning (Shen et al., 2018; Klocek et al., 2019; Serrà et al., 2019; Meyerson and Miikkulainen, 2019), and block code decoding (Nachmani and Wolf, 2019), their application within NLP remains limited.

HNs are effective in language modeling (Suarez, 2017), cross-task cross-language adaptation (Bansal et al., 2020; Üstün et al., 2022), and machine translation (Platanios et al., 2018). Additionally, Üstün et al. (2020) and Mahabadi et al. (2021) leveraged HNs in Transformer architectures (Vaswani et al., 2017) for cross-lingual parsing and multi-task learning, generating adapter (Houlsby et al., 2019) weights while keeping pre-trained language model weights fixed. Mahabadi et al. (2021) addressed the supervised DA setup, where labeled data from the target domain is available.

Our work applies HNs to generate task classifier weights, jointly training the HN and fine-tuning

a language model. We also incorporate example-based adaptation as per Ben-David et al. (2022), a novel HN application within NLP, marking the first instance of HNs deployed in NLP in an example-based fashion. Lastly, we introduce a HN mechanism designed for adaptation to unseen domains.

# 3 Domain Adaptation with Hypernetworks

In this section, we present our HN-based modeling framework for domain adaptation. We present three models in increased order of complexity: We start by generating parameters only for the task classifier in a domain-based manner (Hyper-DN), proceed to example-based classifier parametrization (Hyper-DRF) and, finally, introduce example-based parametrization at both the classifier and the text representation levels (Hyper-PADA).

Throughout this section we use the running example of Table 1. This is a Natural Language Inference (NLI) example from one of our experimental MNLI (Williams et al., 2018) setups. In this task, the model is presented with two sentences, Premise and Hypothesis, and it should decide the relationship of the latter to the former: Entailment, Contradiction or Neutral (see §4).

§3.1 describes the model architectures and their training procedure. We refer the reader to Appendix A.1 for more specific details of the DRF scheme, borrowed from Ben-David et al. (2022). The DRFs are utilized to embed input examples in the semantic space of the source domains, hence supporting example-based classifier parametrization and improved example representation.

## 3.1 Models

**Hyper Domain Name (Hyper-DN)**  Our base model (Figure 3b) unifies a pre-trained T5 language encoder, a classifier (CLS), and a hypernetwork (HN) that generates classifier weights. Hyper-DN feeds the domain name into the HN. As domain names are unknown during test-time inference, we employ an "UNK" token to represent all unknown domains. To familiarize the model with this token, we apply it to an $\alpha$ proportion of training examples during training instead of the domain name. This architecture promotes parameter sharing across domains and optimizes weights for unknown domains at the classifier level.

In Table 1, the test example's premise and hypothesis enter the T5 encoder, while the "UNK" token goes to the HN. In this model, neither domain-name nor example-specific signature is generated.

**Hyper-DRF**  We hypothesize that parameter sharing in the domain level may lead to suboptimal performance. For instance, the sentence pair of our running example is taken from the *Government* domain but is also semantically related to the *Travel* domain. Thus, we present **Hyper-DRF** (Figure 3c), an example-based adaptation architecture, which makes use of domain-related features (DRFs) in addition to the domain name. Importantly, this model may connect the input example to semantic aspects of multiple source domains.

*Hyper-DRF* is a multi-stage multi-task autoregressive model, which first generates a DRF signature for the input example, and then uses this signature as an input to the HN. The HN, in turn, generates the task-classifier (CLS) weights, but, unlike in Hyper-DN, these weights are example-based rather than domain-based. The model is comprised of the following components: **(1)** a T5 encoder-decoder model which generates the DRF signature of the input example in the first stage (*travel: city, area, town, reports, modern* in our running example); **(2)** a (separate) T5 encoder to which the example is fed in the second stage; and **(3)** a HN which is fed with the DRF signature, as generated in the first stage, and generates the weights of the task-classifier (CLS). This CLS is fed with the example representation, as generated by the T5 encoder of (2), to predict the task label.

Below we discuss the training of this model in details. The general scheme is as follows: We first train the T5 encoder-decoder of the first stage ((1) above), and then jointly train the rest of the architecture ((2) and (3) above), which is related to the second stage. For the first training stage we have to assign each input example a DRF signature. In §A.1 we provide the details of how, following Ben-David et al. (2022), the DRF sets of the source training domains are constructed based on the source domain training corpora, and how a DRF signature is comprised for each training example in order to effectively train the DRF signature generator ((1) above). For now, it is sufficient to say that the DRF set for each source domain includes words strongly linked to that domain, and each example's DRF signature is a sequence of these DRFs (words).

During inference, when introduced to an example from an unknown domain, *Hyper-DRF* generates its DRF signature using its trained generator

(T5 encoder-decoder). This way, the signature of a test example may consist of features from the DRF sets of one or more source domains, forming a mixture of semantic properties of these domains. In our running example, while the input sentence pair is from the unknown *Government* domain, the model generates a signature based on the *Travel* and *Slate* domains. Importantly, unlike in Hyper-DN, there is no need in an "UNK" token as input to the HN since the DRF signatures are example-based.

**Hyper-PADA** While Hyper-DRF implements example-based adaptation, parameter-sharing is modeled (apart from the T5 model) only at the classifier level: The language representation (with the T5 encoder) is left untouched. Our final model, **Hyper-PADA**, casts the DRF-based signature generated at the first stage of the model, both as a prompt concatenated to the input example before it is fed to the T5 language encoder, and as an input to the HN.

Specifically, the architecture of *Hyper-PADA* (Figure 3d) is identical to that of Hyper-DRF. At its first stage, which is identical to the first stage of Hyper-DRF, it employs a generative T5 encoder-decoder which learns to generate an example-specific DRF signature for each input example. Then, at its second stage, the DRF signature is used in two ways: (A) unlike in Hyper-DRF, it is concatenated to the input example as a prompt, and the concatenated example is then fed into a T5 encoder, in order to create a new input representation (in Hyper-DRF the original example is fed into the T5 encoder); and (B) as in Hyper-DRF, it is fed to the HN which generates the task-classifier weights. Finally, the input representation constructed in (A) is fed into the classifier generated in (B) in order to yield the task label.

**Training** While some aspects of the selected training protocols are based on development data experiments (§4), we discuss them here in order to provide a complete picture of our framework.

For Hyper-DN, we found it most effective to jointly train the HN and fine-tune the T5 encoder using the task objective. As discussed above, Hyper-DRF and Hyper-PADA are multi-stage models, where the HN (in both models) and the T5 language encoder (in hyper-PADA only) utilize the DRF signature generated in the first stage by the T5 encoder-decoder. Our development data experiments demonstrated significant improvements

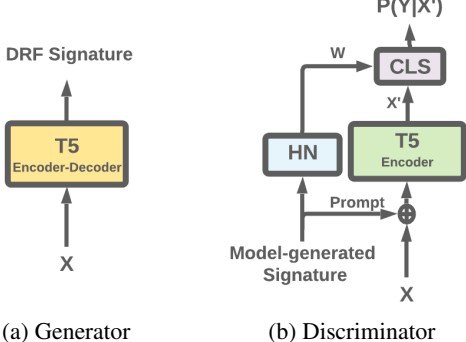

(a) Generator      (b) Discriminator

Figure 2: *Hyper-PADA* training. The generative (T5 encoder-decoder) and discriminative (HN, T5 Ecnoder and CLS) components are trained separately, using source domains examples.

when using one T5 encoder-decoder for the first stage, and a separate T5 encoder for the second stage. Moreover, since the output of the first stage is discrete (a sequence of words), we cannot train all components jointly.

Hence, we train each stage of these models separately, as shown in Figure 2. First, the T5 encoder-decoder is trained to generate the example-based DRF signature. Then, the HN and the (separate) T5 encoder are trained jointly with the task objective.

## 4 Experimental Setup

### 4.1 Tasks, Datasets, and Setups

While our focus is on domain adaptation, the availability of multilingual pre-trained language encoders allows us to consider two setups: (1) Cross-domain transfer (CD); and (2) cross-language cross-domain transfer (CLCD). We consider multi-source adaptation and experiment in a leave-one-out fashion: In every experiment we leave one domain (CD) or one domain/language pair (CLCD) out, and train on the datasets that belong to the other domains (CD) or the datasets that belong to both other domains and other languages (CLCD; neither the target domain nor the target language are represented in the training set).[3]

**Data set sizes** Despite the growing ability to collect massive datasets, obtaining large labeled datasets is still costly and labor-intensive. When addressing a new task with a limited annotation budget, the choice lies between focusing efforts on a single domain or distributing the effort across multiple domains, acquiring fewer examples from

---

[3]URLs of the datasets, implementation details, and hyper-parameter configurations are described in Appendix D.

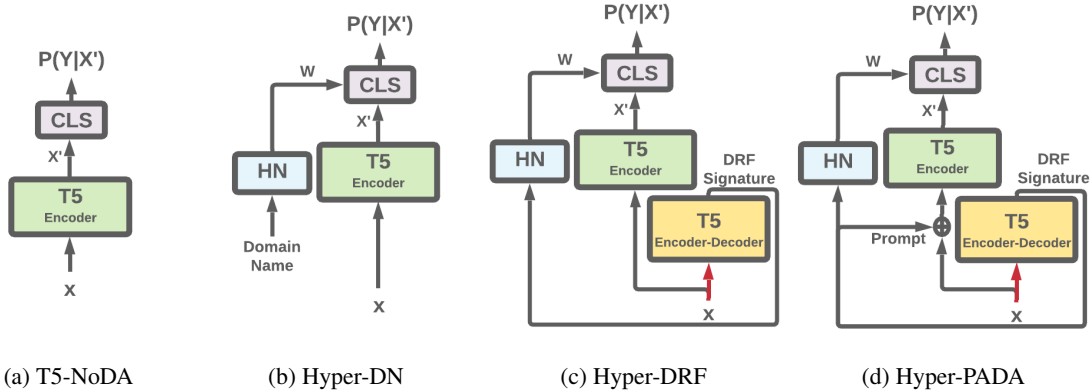

|  (a) T5-NoDA | (b) Hyper-DN | (c) Hyper-DRF | (d) Hyper-PADA |

Figure 3: The four models representing the evolution of our HN-based domain adaptation framework. From left to right: **T5-NoDA** is a standard NLP model comprised of a pre-trained T5 encoder with a classifier on top of it, both are fine-tuned with the downstream task objective. **Hyper-DN** employs an additional hypernetwork (HN), which generates the classifier (CLS) weights given the domain name (or an "UNK" specifier for examples from unknown domains). **Hyper-DRF** and **Hyper-PADA** are multi-stage multi-task models (first-stage inputs are in red, second stage inputs in black), comprised of a T5 encoder-decoder, a separate T5 encoder, a HN and a task classifier (CLS). At the first stage, the T5 encoder-decoder is trained for example-based DRF signature generation (§A.1). At the second stage, the HN and the T5 encoder are jointly trained using the downstream task objective. In Hyper-PADA, the DRF signature of the first stage is applied both for example representation and HN-based classifier parametrization, while in Hyper-DRF it is applied only for the latter purpose. In all HN-based models, our HN is a simple two-layer feed-forward NN (§D).

each while maintaining a similar overall data size. This work explores the latter scenario. To follow the experimental setup presented in previous DA works (Guo et al., 2018; Wright and Augenstein, 2020; Ben-David et al., 2022) and to perform realistic experiments, we hence adjust our multi-domain datasets. In each experiment, we downsample each domain to have several thousand (3K-10K) training examples (with a proportionate development set).

**Cross-domain Transfer (CD) for Natural Language Inference** We experiment with the MNLI dataset (Williams et al., 2018). In this task, each example consists of a premise-hypothesis sentence pair and the relation between the the latter and the former: Entailment, contradiction, or neutral. The corpus consists of ten domains, five of which are split to train, validation, and test sets, while the other five do not have training sets. We experiment with the former five: Fiction (F), Government (G), Slate (S), Telephone (TL), and Travel (TR).

Since the MNLI test sets are not publicly available, we use the validation sets as our test sets and split the training sets to train and validation. Following our above multi-domain setup, we downsample each domain so that in each experiment we have 10,000 training (from all source domains jointly) , 800 validation and about 2000 test exam-

ples (see details in §B).

**Cross-language Cross-domain (CLCD) and Multilingual Cross-domain (CD) Transfer for Sentiment Analysis** We experiment with the task of sentiment classification, using the Websis-CLS-10 dataset (Prettenhofer and Stein, 2010), which consists of Amazon reviews from 4 languages (English (En), Deutsch (De), French (Fr), and Japanese (Jp)) and 3 product domains (Books (B), DVDs (D), and Music (M)).

We perform one set of multilingual cross-domain (CD) generalization experiments and one set of cross-language cross-domain (CLCD) experiments. In the former, we keep the training language fixed and generalize across domains, while in the latter we generalize across both languages and domains. Hence, experimenting in a leave-one-out fashion, in the CLCD setting we focus each time on one domain/language pair. For instance, when the target pair is *English-Books*, we train on the training sets of the *{French, Deutsch, Japanese}* languages and the *{Movies, Music}* domains (a total of 6 sets), and the test set consists of *English* examples from the *Books* domain. Likewise, in the CD setting we keep the language fixed in each experiment, and generalize from two of the domains to the third one. We hence have 12 CLCD experiments (one with

| | Deutsch | | | English | | | French | | | Japanese | | | |
|---|---|---|---|---|---|---|---|---|---|---|---|---|---|
| | **B** | **D** | **M** | **B** | **D** | **M** | **B** | **D** | **M** | **B** | **D** | **M** | **Avg** |
| **T5-NoDA** | 77.1 | 75.8 | 63.9 | 78.4 | 78.8 | 64.5 | 83.0 | 82.6 | 75.1 | 61.5 | 79.9 | 79.7 | 75.0 |
| **T5-MoE-Ind-Avg** | 81.9 | 76.6 | 79.6 | 86.0 | 81.2 | 81.6 | 85.0 | 84.9 | 77.2 | 82.2 | 83.6 | 82.0 | 81.8 |
| **T5-MoE-Ind-Attn** | 82.1 | 76.2 | 79.6 | 86.0 | 82.6 | 81.7 | 84.6 | 84.6 | 77.4 | 81.8 | 82.2 | 82.9 | 81.8 |
| **T5-MoE-Avg** | 81.6 | 76.7 | 79.0 | 85.7 | 81.5 | 81.6 | 85.0 | 84.8 | 77.0 | 82.2 | 83.4 | 81.9 | 81.7 |
| **T5-IRM** | 71.2 | 70.2 | 75.8 | 80.8 | 72.5 | 73.0 | 82.3 | 80.6 | 78.4 | 75.5 | 75.8 | 78.4 | 76.2 |
| **T5-DANN** | 82.1 | 77.8 | 80.8 | 84.6 | 78.8 | 79.0 | 84.2 | 82.6 | 77.2 | 68.7 | 78.8 | 81.6 | 79.7 |
| **PADA** | 57.7 | 74.8 | 74.2 | 71.8 | 75.9 | 78.8 | 81.8 | 82.0 | 76.8 | 77.2 | 78.8 | 80.0 | 75.8 |
| **Hyper-DN** | **86.2** | 80.8 | 84.4 | 85.6 | 84.2 | 83.4 | 86.5 | 84.5 | 81.6 | 81.3 | 82.0 | 83.2 | 83.7 |
| **Hyper-DRF** | 85.9 | 81.2 | 84.6 | 86.4 | 84.0 | 83.9 | 85.7 | 85.5 | 81.4 | 82.2 | 82.0 | **83.9** | 83.9 |
| **Hyper-PADA** | 85.7‡◇+ | **81.8**♣‡◇+ | **85.0**♣‡+ | 86.0‡◇ | **84.4**♣‡◇+ | **85.1**♣◇+ | **86.6**♣‡◇+ | **85.9**‡◇+ | **81.8**♣‡◇+ | **83.9**‡◇+ | **83.9**♣◇+ | 83.8‡◇+ | **84.5** |
| **Upper-bound** | 86.7 | 83.8 | 86.4 | 88.7 | 85.9 | 86.9 | 87.9 | 87.3 | 83.9 | 84.4 | 86.4 | 86.9 | 86.3 |

Table 2: CLCD sentiment classification accuracy. The statistical significance of the Hyper-PADA results (with the McNemar paired test for labeling disagreements (Gillick and Cox, 1989), $p < 0.05$) is denoted with: ♣ (vs. the best of Hyper-DN and Hyper-DRF), + (vs. the best domain expert model), ◇ (vs. the best domain robustness model), and ‡ (vs. PADA (example-based adaptation)).

each language/domain pair as target) and 12 CD experiments (for each language we perform one experiment with each domain as target). Following our above multi-domain setup, we downsample each language-domain pair so that each experiment includes 3000 train, 600 validation and 2000 test examples (see details in §B).

## 4.2 Models and Baselines

We compare our HN based models (**Hyper-DN**, **Hyper-DRF**, and **Hyper-PADA**) to models from three families (see §1): (a) *domain expert models* that do not share information across domains: A model trained on the source domains and applied to the target domain with no adaptation effort (*T5-NoDA*); and three mixture of domain-specific expert models (Wright and Augenstein, 2020), where a designated model is trained on each source domain, and test decisions are made through voting between the predictions of these models (*T5-MoE-Ind-Avg*, *T5-MoE-Ind-Attn*, and *T5-MoE-Avg*); (b) *domain robustness models*, targeting generalization to unknown distributions through objectives that favor robustness over specification (*T5-DANN* (Ganin and Lempitsky, 2015) and *T5-IRM* (Arjovsky et al., 2019)); and (c) *example-based multi-source adaptation* through prompt learning (*PADA*, the SOTA model for our setup).

Below we briefly discuss each of these models. All models, except from T5-MoE, are trained on a concatenation of the source domains training sets.

**(a.1) T5-No-Domain-Adaptation (T5-NoDA)**. A model consisting of a task classifier on top of a T5 encoder. The entire architecture is fine-tuned on the downstream task (see Figure 3a).

**(a.2-4) T5-Mixture-of-Experts (T5-MoE-Ind-Avg, T5-MoE-Ind-Attn, T5-MoE-Avg)**. Our implementation of the *Independent Avg*, *Independent*

*Fine Tune*, and *MoE Avg* models presented by Wright and Augenstein (2020)[4]. For **T5-MoE-Ind-Avg**, we fine-tune an expert model (with the same architecture as *T5-NoDA*) on the training data from each source domain. At inference, we average the class probabilities of all experts, and the class with the maximal probability is selected.

For **T5-MoE-Ind-Attn**, we train an expert model for each source domain. Then, in order to find the optimal weighted expert combination, we perform a randomized grid search on our (source domain) development set. Finally, **T5-MoE-Avg** is similar to *T5-MoE-Ind-Avg* except that we also include a general-domain expert, identical to *T5-NoDA*, in the expert committee.

**(b.1) T5-Invariant-Risk-Minimization (T5-IRM)**. Using the same architecture as *T5-NoDA*, but with an objective term that penalizes representations with different optimal classifiers across domains.

**(b.2) T5-Domain-Adversarial-Network (T5-DAN)**. An expert with the same architecture as *T5-NoDA*, but with an additional adversarial domain classifier head (fed by the T5 encoder) which facilitates domain invariant representations.

**(c.1) PADA**. A T5 encoder-decoder that is fed with each example and generates its DRF signature. The example is then appended with this signature as a prompt, fed again to the T5 encoder and the resulting representation is fed into the task classifier. We follow the implementation and training details from (Ben-David et al., 2022).

For each setup we also report an upper-bound: The performance of the model trained on the training sets from all source domains (or source language/domain pairs in CLCD) including that of

---

[4]For the MoE models, we follow the naming conventions of Wright and Augenstein (2020).

the target, when applied to the target domain's (or language/domain pair in CLCD) test set.

# 5 Results

Table 2 and Figure 4 present sentiment classification accuracy results for CLCD and CD transfer, respectively (12 settings each), while Table 3 reports Macro-F1 results for MNLI in 5 CD settings. We report accuracy or F1 results for each setting, as well as the average performance across settings. Finally, we report statistical significance following the guidelines at Dror et al. (2018), comparing Hyper-PADA to the best performing model in each of the three baseline groups discussed in §4: (a) domain expert models (T5-NoDA and T5-MoE) ;(b) domain robustness models (T5-DANN and T5-IRM) and (c) example-based adaptation (PADA). We also report whether the improvement of Hyper-PADA over the simpler HN-based models, Hyper-DN and Hyper-DRF, is significant.

Our results clearly demonstrate the superiority of Hyper-PADA and the simpler HN-based models. Specifically, Hyper-PADA outperforms all baseline models (i.e. models that do not involve hypernetwork modeling, denoted bellow as non-HN models) in 11 of 12 CLCD settings, in 8 of 12 CD sentiment settings, and in all 5 CD MNLI settings, with an average improvement of 2.7%, 3.9% and 3.4% over the best performing baseline in each of the settings, respectively. Another impressive result is the gap between Hyper-PADA and the T5-NoDA model, which does not perform adaptation: Hyper-PADA outperforms this model by 9.5%, 8.4% and 14.8% in CLCD and CD sentiment classification and CD MNLI, respectively.

Hyper-DN and Hyper-DRF are also superior to all non-HN models across settings (Hyper-DRF in 10 CLCD sentiment settings, in 7 CD sentiment settings and in 2 CD MNLI settings, as well as on average in all three tasks; Hyper-DN in 7 CLCD sentiment settings, in 6 CD sentiment settings, and in 2 CD MNLI settings, as well as on average in all three tasks). It is also interesting to note that the best performing baselines (non-HN models) are different in the three tasks: While T5-MoE (group (a) of domain expert baselines) and T5-DANN (group (b) of domain robustness baselines) are strong in CLCD sentiment classification, PADA (group (c) of example-based adaptation baselines) is the strongest baseline for CD MNLI (in CD sentiment classification the average perfor-

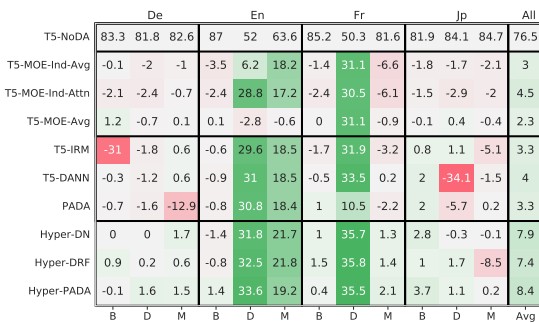

| | De | | | En | | | Fr | | | Jp | | | All |
|---|---|---|---|---|---|---|---|---|---|---|---|---|---|
| T5-NoDA | 83.3 | 81.8 | 82.6 | 87 | 52 | 63.6 | 85.2 | 50.3 | 81.6 | 81.9 | 84.1 | 84.7 | 76.5 |
| T5-MOE-Ind-Avg | -0.1 | -2 | -1 | -3.5 | 6.2 | 18.2 | -1.4 | 31.1 | -6.6 | -1.8 | -1.7 | -2.1 | 3 |
| T5-MOE-Ind-Attn | -2.1 | -2.4 | -0.7 | -2.4 | 28.8 | 17.2 | -2.4 | 30.5 | -6.1 | -1.5 | -2.9 | -2 | 4.5 |
| T5-MOE-Avg | 1.2 | -0.7 | 0.1 | 0.1 | -2.8 | -0.6 | 0 | 31.1 | -0.9 | -0.1 | 0.4 | -0.4 | 2.3 |
| T5-IRM | -31 | -1.8 | 0.6 | -0.6 | 29.6 | 18.5 | -1.7 | 31.9 | -3.2 | 0.8 | 1.1 | -5.1 | 3.3 |
| T5-DANN | -0.3 | -1.2 | 0.6 | -0.9 | 31 | 18.5 | -0.5 | 33.5 | 0.2 | 2 | -34.1 | -1.5 | 4 |
| PADA | -0.7 | -1.6 | -12.9 | -0.8 | 30.8 | 18.4 | 1 | 10.5 | -2.2 | 2 | -5.7 | 0.2 | 3.3 |
| Hyper-DN | 0 | 0 | 1.7 | -1.4 | 31.8 | 21.7 | 1 | 35.7 | 1.3 | 2.8 | -0.3 | -0.1 | 7.9 |
| Hyper-DRF | 0.9 | 0.2 | 0.6 | -0.8 | 32.5 | 21.8 | 1.5 | 35.8 | 1.4 | 1 | 1.7 | -8.5 | 7.4 |
| Hyper-PADA | -0.1 | 1.6 | 1.5 | 1.4 | 33.6 | 19.2 | 0.4 | 35.5 | 2.1 | 3.7 | 1.1 | 0.2 | 8.4 |
| | B | D | M | B | D | M | B | D | M | B | D | M | Avg |

Figure 4: Accuracy improvements over T5-NoDA, in cross-domain (CD) generalization for four languages: German, English, French, and Japanese. From the 28 out of 36 settings where Hyper-PADA outperforms the best model in each of the baselines groups, in 23 cases the difference is significant (following Table 2 protocol).

mance of all baselines is within a 1% regime). This observation is related to another finding: Using the DRF-signature as a prompt in order to improve the example representation is more effective in CD MNLI – which is indicated both by the strong performance of PADA and the 3.1 F1 gap between Hyper-PADA and Hyper-DRF – than in CLCD and CD sentiment classification – which is indicated both by the weaker PADA performance and by the 0.6% (CLCD) and 1% (CD) accuracy gaps between Hyper-PADA and Hyper-DRF.

These findings support our modeling considerations: (1) integrating HNs into OOD generalization modeling (as the HN-based models strongly outperform the baselines); and (2) integrating DRF signature learning into the modeling framework, both as input to the HN (Hyper-DRF and Hyper-PADA) and as means of improving example representation (Hyper-PADA). In Appendix C we present additional analysis: (a) Hyper-PADA's performance on seen domains; (b) model performance as a function of the training set size; (c) more efficient variations of the Hyper-PADA architecture.

To assess the significance of DA approaches in the era of large language models (LLMs), we executed an ablation study, employing the GPT-3 model (Davinci-003) in a few-shot learning setup, despite the potential data leakage due to the model's likely prior exposure to our test sets (Aiyappa et al., 2023). In this setup, the GPT-3 model was provided a comprehensive task description, paired with an example for each label from every source domain; for instance, given k source domains and m labels, the prompt incorporated k·m few-shot examples. Subsequently, we tested

| | F | G | S | TL | TR | Avg |
|---|---|---|---|---|---|---|
| **T5-NoDA** | 58.2 | 66.0 | 60.2 | 74.3 | 69.1 | 65.6 |
| **T5-MoE-Ind-Avg** | 55.6 | 65.3 | 57.7 | 58.1 | 64.3 | 60.2 |
| **T5-MoE-Ind-Attn** | 55.6 | 64.6 | 59.1 | 59.3 | 64.5 | 60.6 |
| **T5-MoE-Avg** | 56.7 | 66.4 | 60.0 | 67.9 | 65.4 | 63.3 |
| **T5-IRM** | 51.1 | 64.6 | 51.7 | 54.7 | 64.5 | 57.3 |
| **T5-DANN** | 72.1 | 76.9 | 65.7 | 74.8 | 76.1 | 73.1 |
| **PADA** | 76.7 | 79.6 | 75.3 | 78.1 | 75.2 | 77.0 |
| **Hyper-DN** | 74.5 | 81.2 | 74.9 | 76.7 | 79.8 | 77.4 |
| **Hyper DRF** | 75.3 | 82.3 | 73.8 | 76.3 | 78.7 | 77.3 |
| **Hyper PADA** | **79.0**♣‡◇+ | **84.1**♣‡◇+ | **78.2**♣‡◇+ | **79.8**♣◇+ | **81.1**‡ | **80.4** |
| **Upper-bound** | 80.2 | 85.8 | 77.9 | 81.5 | 83.4 | 81.8 |

Table 3: Cross-domain MNLI results (Macro-F1). The statistical significance of Hyper-PADA vs. the best baseline from each group (with the Bootstrap test, $p < 0.05$) is denoted similarly to Table 2.

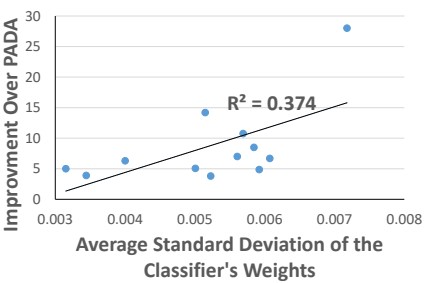

Figure 5: Correlation between the diversity of the example-based classifier weights generated by Hyper-PADA, and the improvement of this model over PADA in CLCD sentiment classification. Each point in the graph represents a target domain. To estimate the SD, we calculated the SD of each of the weights of the HNs generated for the test examples of this domain, and reported the average. The Spearman Correlation is 0.475. For CD sentiment classification, the corresponding numbers are 0.539 and 0.175, for Pearson and Spearman correlations, respectively (not shown in the graph).

this 175B parameter model on our CD MNLI and CLCD sentiment analysis tasks. For CD MNLI, GPT-3 yielded an average F1 score of 75.1, a decline of 5.3% relative to Hyper-PADA. Conversely, in the sentiment analysis task GPT-3 achieved an average accuracy of 90.1, outperforming Hyper-PADA by 5.6%. Despite any potential impact of test data leakage, the results indicate that LLMs still have some distance to cover before their discriminative abilities can outstrip fine-tuned models. These findings underscore the continuous need for in-depth DA methods research. See Appendix D.2 for experiment details.

**Importance of Diversity in Generated Weights**
To demonstrate the impact of example-based classifier parametrization, Figure 5 plots the diversity of the example-based classifier weights as generated by Hyper-PADA vs. the improvement of Hyper-PADA over PADA in the CLCD sentiment classification settings. We choose to compare these models because both of them use the self-generated signature for improved example representation, but only Hyper-PADA uses it for classifier parametrization. To estimate the diversity of the weights generated by the HN in a given test domain, we first measure the standard deviation of each weight generated by the HN across the test examples of that test domain. We then average the SDs of these weights and report the resulting number as the diversity of the HN-generated weights in the test domain. We repeat this process for each test domain. The relatively high correlations between the two measures is an encouraging indication, suggesting the potential importance of example-based parametrization for improved task performance.

## 6 Discussion

We presented a Hypernetwork-based framework for example-based domain adaptation, designed for multi-source adaptation to unseen domains. Our framework provides several novelties: (a) the application of hypernetworks to unsupervised domain adaptation and any domain adaptation in NLP; (b) the application of hypernetworks in example-based manner (which is novel at least in NLP, to the best of our knowledge); (c) the generation of example-based classifier weights, based on a learned signature which embeds the input example in the semantic space spanned by the source domains; and (d) the integration of all the above with an example representation mechanism that is based on the learned signature. While the concept of DRF signatures stems from Ben-David et al. (2022), the aforementioned innovations are unique to our work. Comprehensive experiments across 2 tasks, 4 languages, 8 domains, and 29 adaptation settings underline our framework's superiority over previous methodologies and the value of our modeling choices.

## 7 Limitations

**Extending beyond sequence classification** Although our experimental setup is broad and extensive, the tasks we considered are limited to sentence-level classification tasks. However, there are many other NLP tasks that present challenging

out-of-distribution scenarios. Since it is not trivial to extend HNs effectively to token-level classification or text generation, we would like to address such cases in future work. Ultimately, our goal is to shape our methodology to the level that NLP technology becomes available to as many textual domains as possible, with minimum data annotation and collection efforts.

**Utilizing large models**  Our modeling solution consists of a large pretrained language model. While one could apply the same method using smaller models (available today), it might lead to an unsatisfying performance level compared to the ones reported in this work.

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

## A Additional Background

### A.1 Domain Related Features (DRFs)

In order to perform example-based domain adaptation, the first stage of the Hyper-DRF and Hyper-PADA models maps each input example into a sequence of Domain Related Features (DRFs). Selecting the DRF sets of the source domains is hence crucial for these models, as they should allow the models to map input examples to the semantic space of the source domains. Since a key goal of example-based adaptation is to account for soft domain boundaries, it is important that the DRF set of each source domain should reflect both the unique semantic aspects of this domain and the aspects it shares with other source domains.

To achieve these goals, we follow the definitions, selection, and annotation processes in Ben-David et al. (2022). For completeness, we briefly describe these ideas here.[5]

**DRF Set Construction** Let $S$ be the set of all source domains, and $S_j \in S$ the domain for which we construct the DRF set. We perform the following selection process, considering all the training data from the participating source domains. First, we define the domain label of a sentence to be 1 if the sentence is from $S_j$ and 0 otherwise. We then look for the top $l$ words with the highest *mutual information (MI)* with the 0/1 labels. Then, since MI could indicate association with each of the labels (related to the domain (1) or not (0)), and we are interested only in words associated with the domain, we select only words that meet the criterion:

$$\frac{C_{S \setminus S_j}(w)}{C_{S_j}(w)} \le \rho, \ C_{S_j}(w) > 0$$

Where $C_{S \setminus S_j}(w)$ is the count of the word $w$ in all of the source domains except $S_j$, $C_{S_j}(w)$ is the word count in $S_j$ and $\rho$ is a domain-specificity parameter: The smaller it is, the stronger is the association. The DRF set of $S_j$ is denoted with $R_j$.

**Annotating DRF-based Signatures for Training** In order to train the DRF signature generator of

---

[5]We also implemented an alternative approach which extracts DRF sets based on a TF-IDF criterion. Yet, we noticed that the extracted DRFs are very similar to the ones extracted by the method of Ben-David et al. (2022), which we use for the main results of this paper, and so are the downstream task performances. For instance, in the MNLI task, the average performance differences between implementations with the two DRF selection methods, for Hyper-DRF and Hyper-PADA are 0.1% and 0.6%, respectively.

---

| |
|---|
| **Sentence.** *This documentary is poorly produced, has terrible sound quality and stereotypical "life affirming" stories. There was nothing in here to support Wal-Mart, their business practices or their philosophy.* |
| **Domain.** *DVD.* |
| **Label.** *Negative.* |
| **DRF Signature.** *music: history, rock, sound, story* |

Table 4: An example of Hyper-DRF and Hyper-PADA application to a sentiment classification example. The source domains are *Books*, and *Music*. Generated DRF features from the *Books* and *Music* domains are in blue and green, respectively.

Hyper-DRF and Hyper-PADA we have to construct a DRF signature for each training example. Our goal in this process is to match each training example with those DRFs in its domain's DRF set that are most representative of its semantics. We do this in an automatic manner.

Let $w_1, ... w_n$ be the tokens of a sentence $x$ from the domain $S_j$. Each DRF $r_j \in R_j$ is assigned with the following score:

$$score(r_j, \{w_1, ... w_n\} \in x) = \min_{i=1,...n} \{s(r_j, w_i)\}$$

$$s(r_j, w_i) = \|\Phi(r_j) - \Phi(w_i)\|_2^2$$

where $\Phi(x)$ is the embedding of $x$ in the pre-trained embedding layer of an off-the-shelf BERT model. Then, let $T_1, ..., T_k$ be the $k$ DRFs with the lowest scores and $D$ the domain name. We define the DRF signature of $x$ to be the following string: "$D : T_1, ..., T_k$".

To summarize, we utilize this annotation only during training, as a training signal for the DRF signature generator (in stage 1 of both Hyper-DRF and Hyper-PADA).

Tables 1 (main paper) and 4 provide MNLI and sentiment classification examples and their DRF signatures, as generated by Hyper-PADA and Hyper-DRF in a specific adaptation setup.

## B Dataset Sizes

Table 5 presents the number of training, development and test examples from each domain. Notice that since we consider multiple training domains in each of our experiments, the number of training and development examples in our experiments are an aggregation of the numbers shown in the table. For example, in the CLCD sentiment analysis task, when we test on the English DVD domain, we use 3000 training examples, 600 development examples and 2000 test examples. In each experiment,

| Sentiment Analysis (En, De, Fr, Jp) | | | |
|---|---|---|---|
| **Domain** | **Training (src)** | **Dev (src)** | **Test (trg)** |
| **Books (B)** | 500 | 100 | 2000 |
| **DVD (D)** | 500 | 100 | 2000 |
| **Music (M)** | 500 | 100 | 2000 |
| MNLI (En) | | | |
| **Domain** | **Training (src)** | **Dev (src)** | **Test (trg)** |
| **Fiction (F)** | 2500 | 200 | 1,973 |
| **Government (G)** | 2500 | 200 | 1,945 |
| **Slate (SL)** | 2500 | 200 | 1,955 |
| **Telephone(TL)** | 2500 | 200 | 1,966 |
| **Travel (TR)** | 2500 | 200 | 1,976 |

Table 5: The number of examples in each domain (and language) of our two tasks. We denote the examples used when a domain is included as a source domain (src), and when it is the target domain (trg). For sentiment we present the number of examples in a single language, while there are four different languages - English (En), Deutsch (De), French (Fr), and Japanese (Jp), each with the same number of examples per domain.

the source domains development sets are used in order to select the hyper-parameters of the models.

## C  Ablation Analysis

**Training Size Effect**  Our experiments focus on scenarios that are both low-resource and domain adaptation, as the combination of the two yields a very challenging, yet realistic, generalization setup (Landeiro and Culotta, 2018; Calderon et al., 2022). Yet, it is also essential to assess the impact of our modeling approach across training sets of various sizes, including cases where labeled data is abundant. Hence, we next turn to evaluate Hyper-PADA as it compares to other baselines, T5-NoDA, T5-DANN and T5-IRM, across multiple subsets of the training data available for our tasks (sentiment analysis and MNLI). We experiment with the following subset sizes: 10%-100% (in 10% steps) for the CLCD setting (sentiment analysis); and 1%-5% (in 1% steps) and 10%-100% (in 10% steps) for the CD setting of MNLI. For each experiment, we sample a subset of the corresponding percentage from the training examples of each of the source domains and use the same test and validation sets across all experiments.

Figure 6 summarizes our results. Figure 6a presents sentiment classification results for the CLCD transfer, including subsets ranging from 10% to 100% (for a total of 10 subset points). Figure 6b presents results for MNLI in the CD transfer, with subsets ranging from 1% to 20% (with 7 subset points) and Figure 6c focuses on the MNLI

|  | **Sentiment CLCD** | **Sentiment CD** | **MNLI** |
|---|---|---|---|
| **T5-NoDA** | 78.7 | 82.0 | 65.2 |
| **T5-MoE-Ind-Avg** | 83.8 | 80.4 | 59.0 |
| **T5-MoE-Ind-Attn** | 84.7 | 84.0 | 59.9 |
| **T5-MoE-Avg** | 83.6 | 80.0 | 61.9 |
| **T5-IRM** | 77.1 | 81.8 | 57.1 |
| **T5-DANN** | 81.3 | 79.0 | 72.2 |
| **PADA** | 78.6 | 83.0 | 77.1 |
| **Hyper-DN** | 86.7 | **85.7** | 77.1 |
| **Hyper-DRF** | 86.8 | 85.1 | 77.7 |
| **Hyper-PADA** | **87.5** | 85.5 | **80.6** |

Table 6: Seen domains results. HN-based methods are superior.

subsets corresponding to subsets larger than 30% (with 8 subset points). Each point in the presented graphs presents the average performance across all settings. For instance, the point corresponding to 10% in CLCD sentiment analysis presents the average performance across all CLCD settings (12 overall). Accordingly, each of the 12 settings uses 10% of the training examples of the corresponding source domains (we sample a subset of the 10% from each source domain).

For sentiment classification, Figure 6a presents a clear and stable trend across all subsets: Hyper-PADA is superior to all three baselines. The performance gap between the methods is more significant in low-resource scenarios (smaller training subsets). Furthermore, while Hyper-PADA's advantage decreases as the labeled training size grows, it still performs better than its baselines across all training set sizes.

For MNLI, when considering up to 20% of the data (more than 60K training examples), Hyper-PADA significantly outperforms all three baseline, as demonstrated in Figure 6b. For larger subsets (more than 30%, Figure 6c), Hyper-PADA, T5-DANN and T5-NoDA demonstrate compatible performance, while T5-IRM reaches significantly lower results. We note that for subsets of 30% of the MNLI data, the models train on more than 22.5K examples from each source domain (for a total of 90K training examples), which seems to be enough to overcome the OOD effect. For comparison, the 100% subsets of the CLCD sentiment analysis dataset contain 12K examples.

**Evaluating Performance on Seen Domains**  In this paper, we put a strong emphasis on the performance of an algorithm on unseen target domains. Our main reasoning is that compared to the limited number of known source domains, there is potentially an unlimited number of unknown tar-

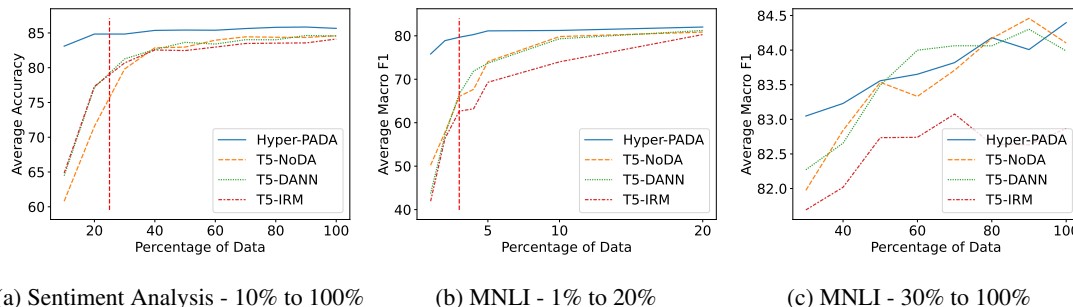

| | | | |
|---|---|---|---|
| (a) Sentiment Analysis - 10% to 100% | (b) MNLI - 1% to 20% | (c) MNLI - 30% to 100% |

Figure 6: The performances of *Hyper-PADA* and *T5-NoDA* on training subsets of different size. The red vertical dashed lines present the training subset size in our main experimental setup.

| #T5 | Model Size | F | G | S | TL | TR | Avg |
|---|---|---|---|---|---|---|---|
| 2 | Small | 69.3 | 75.6 | 65.6 | 69.1 | 71.7 | 70.3 |
| 1 | Small | 67.1 | 71.9 | 61.6 | 67.6 | 67.0 | 67.0 |
| 2 | Base | 79.0 | 84.1 | 78.2 | 79.8 | 81.1 | 80.4 |
| 1 | Base | 78.7 | 84.1 | 76.8 | 78.1 | 80.7 | 79.7 |
| 2 | Large | 85.4 | 88.1 | 83.7 | 85.8 | 88.1 | 86.2 |
| 1 | Large | 85.6 | 88.1 | 79.8 | 85.9 | 86.8 | 85.2 |

Table 7: The results of Hyper-PADA using one or two T5 models. The two models version is the one described in the main paper.

| #HN Layers | #Params | F | G | S | TL | TR | Avg |
|---|---|---|---|---|---|---|---|
| 1 | 2.4M | 79.2 | 83.7 | 76.5 | 80.2 | 79.1 | 79.7 |
| 2 | 3.0M | 79.0 | 84.1 | 78.2 | 79.8 | 81.1 | 80.4 |
| 3 | 3.5M | 77.8 | 83.4 | 77.7 | 79.5 | 80.9 | 79.9 |

Table 8: MNLI F1 results of Hyper-PADA with different number of fully connected layers (1,2,3). For each configuration, we present the number of corresponding HN parameters.

get domains, which the algorithm may encounter in future tests. Still, it is essential to verify that our algorithms do not sacrifice their source domain performance in order to achieve out-of-distribution generalization. Hence, We next measure the performance on the source domains in each experiment by calculating the F1 score (MNLI) or accuracy (sentiment classification) across all development examples from the source domains. In each experiment, we calculate the relevant metric on each source domain's validation set. Then, we average the results of each domain across all runs.

Table 6 reports our results, demonstrating the superiority of our models on seen domains. The HN models are superior in all the setups, with *Hyper-PADA* outperforming all models for sentiment CLCD and MNLI setups and is the second best model in the sentiment CD setup, where it is slightly outperformed by Hyper-DN.

**Dual vs. Solo T5 Model Performance** In the architectures of both Hyper-PADA and Hyper-DRF, we employed two distinct T5 models. One served as a signature generator, while the other, trained from scratch, functioned as an encoder for the discriminative component (refer to Figure 2b in the main paper). As an alternative, one could consider using a single T5 model to perform both roles. In this approach, the training regimen alternates between signature generation and classification tasks (mediated by the hypernetwork). Each training example stands a 20% probability of being used for generation. As evidenced in Table 7, the dual T5 model setup consistently delivers superior performance across all model sizes compared to the single-model approach.

**HN Architectures Variants** We subsequently assessed the best configuration for the HN, explicitly focusing on the number of layers within the HN. The main paper discussed the results of an HN with a single layer responsible for generating classifier weights. In this analysis, we experimented with varying layer counts: 1, 2, and 3, incorporating a ReLU activation after each layer. The findings are presented in Table 8. While the single-layer setup isn't optimal, adding more layers doesn't offer a substantial improvement.

# D Implementation Details

## D.1 URLs of Code and Data

- **Our Code Repository** - our official code repository will be published upon acceptance. In addition, we attach to this submission a zip folder that contains our anonymized code source.

- **HuggingFace** (Wolf et al., 2020) - code and pretrained weights for the T5 model and tok-

- **MNLI dataset** - The natural language inference data experimented with within this paper. `https://cims.nyu.edu/~sbowman/multinli/`

- **Websis-CLS-10 dataset** - The multilingual multi-domain dataset which is experimented with within this paper. `https://zenodo.org/record/3251672#.YdQiIWhBwQ8`

## D.2 Text classification with GPT3

This section elaborates on the methodology employed for text classification with GPT3 within our DA experiments, and provides insight into the implementation process of GPT3 prompting. Our focus is on conducting DA from various source domains to unknown target domains. Given this scenario, a zero-shot GPT3 setup naturally aligns with our paradigm, albeit assuming that the original GPT3 training phase is devoid of data contamination. However, this approach is generally less effective than the few-shot GPT3 alternative.

To incorporate the few-shot approach of GPT-3 while adhering to our experimental protocol, we limit the model to employ only instances from the source domain within each experiment. Our preliminary studies showed enhanced performance when task instructions and specifics were added to assist the model in comprehending the challenge it faces.

Thus, we augment our prompt with task-related knowledge, instructions, and examples from multiple domains. These examples feature one example for each label from every source domain. It's worth noting that while all test examples within a domain-shift experiment employ an identical prompt, we modify the prompt for each distinct domain-shift of the same task, given it incorporates examples from varied domains.

For transparency and reproducibility, in Table 10 we provide the exact prompt we designed for the MNLI experiment. As part of this process, we set the temperature parameter to zero. Similarly, our CLCD sentiment analysis prompt is designed following these same guidelines.

## D.3 Hyperparameter Different Choices

For all the pre-trained models we use the *Huggingface* Transformers library (Wolf et al., 2020). For the T5 model we use the T5-base model (Raffel et al., 2020) for MNLI, and the MT5-base model

|  | MNLI | | Sentiment | |
| --- | --- | --- | --- | --- |
| **Model** | Train | Total | Train | Total |
| **No-DA** | 110 | 110 | 277 | 277 |
| **MOE-Ind** | 439 | 439 | 1662 | 1662 |
| **MOE-Avg** | 548 | 548 | 1939 | 1939 |
| **IRM** | 110 | 110 | 277 | 277 |
| **DANN** | 110 | 110 | 277 | 277 |
| **PADA** | 333 | 442 | 859 | 1027 |
| **Hyper-DN** | 112 | 221 | 280 | 447 |
| **Hyper-DRF** | 335 | 444 | 862 | 1030 |
| **Hyper-PADA** | 335 | 444 | 862 | 1030 |

Table 9: The number in millions of parameters in each model. MOE-IND represents both MOE-IND-Avg and MOE-IND-Attn since the difference is negligible.

(Xue et al., 2021) for sentiment classification. For contextual representation of the HN input (domain name or "UNK' in Hyper-DN, DRF signature in Hyper-DRF and Hyper-PADA), we use the BERT-base-uncased and the mBERT-based-uncased models, for MNLI and sentiment classification, respectively.

We choose $\rho = 1.5$ for the DRF set construction process. In the DRF signature annotation process, we take the $k = 5$ most associated DRFs for each input example. When generating the signature (in Hyper-DRF and Hyper-PADA) we employ the Diverse Beam Search algorithm (Vijayakumar et al., 2016) with the T5 decoder, using the following parameters: 5 sequences, with a beam size of 5, a 5 beams group and a diversity penalty of 0.1.

The HN consists of two linear layers of the same input and output dimensions ($1 \times 768$), each of which is followed by a ReLU activation layer. The output of the second layer is fed into two parallel linear layers, one to predict the weights of the linear classifier (a $2 \times 768$ matrix), and one to predict its bias (a $1 \times 2$ vector). For task classification, we feed the linear classifier (CLS) with the average of the encoder token representations.

Generative models are trained for 3 epochs and discriminative models for 5 epochs. We use the Cross Entropy loss for all models, optimized with the ADAM optimizer (Kingma and Ba, 2015), a batch size of 16, and a learning rate of $5 * 10^-6$. We limit the number of input tokens to 128.

### D.3.1 Computing Infrastructure and Runtime

All experiments were performed on a single Nvidia Quadro RTX 6000 GPU, with 4608 cores, 24 GB GPU memory, 12 CPU cores and 125 GB RAM. For a single CLCD sentiment analysis experiment

Table 10: GPT3 prompt for CD MNLI.

```
## Task:
Classify the following sentence pair,
the 'premise' and 'hypothesis', into
three categories: Entailment,
Contradiction, and Neutral. The
premise begins with 'First: ', the
hypothesis begins with 'Second: ',
and they are seperated with '  '.

## Guidelines:

1. **Entailment**: If the premise is
true, the hypothesis is definitely
true.
2. **Contradiction**: If the premise
is true, the hypothesis is definitely
false.
3. **Neutral**: The premise doesn't
definitively confirm or refute the
hypothesis.

## Tips:

- Understand the context of both
sentences before deciding.
- Look for key words or phrases that
indicate the relationship.
- Don't make assumptions based on
outside knowledge. The premise alone
should dictate your decision.
- If a word has different meanings
in the premise and hypothesis, it
can change the relationship.
- Continuously learn from feedback
and ask for clarification when
needed.

## Example 1:
{src_domain_0_entail]}
Answer:  Entailment

## Example 2:
{src_domain_0_contradict]}
Answer:  Contradiction

## Example 3:
{src_domain_0_neutral}
Answer: Neutral

...
...

## Example 11:
{src_domain_4_contradict}
Answer: Contradiction

## Example 12:
{src_domain_4_neutral}
Answer: Neutral

Now your turn
## Example 13:
{example}
Answer:
```

with Hyper-DN, we measured a runtime of 5 minutes, which corresponds to a single cell in Table 2 (in the Hyper-DN row). Respectively, for a single CD MNLI experiment, we measured a runtime of 12 minutes for Hyper-DN, corresponding to a single cell in Table 3. For Hyper-PADA and Hyper-DRF, we measured a runtime of 20 minutes for a single CLCD sentiment analysis experiment, and 45 minutes for a single MNLI experiment (corresponding to a single cell in Table 2 and a single cell in Table 3 respectively). In table 9 we present the number of parameters in each of the models and baselines used in this paper. While our model has a large number of parameter due to the use of a T5 encoder-decoder and a separate T5 encoder, other methods use a significantly larger (T5-MOE versions) or a comparable number (PADA) while reaching lower results.