# OpenReview forum: "Example-based Hypernetworks for Multi-source Adaptation to Unseen Domains"
_EMNLP/2023/Conference — EMNLP 2023 Findings_

### Official Review · Reviewer_suhr · 2023-08-05

**Typos Grammar Style And Presentation Improvements:** None.
**Soundness:** 2

**Excitement:**

3: Ambivalent: It has merits (e.g., it reports state-of-the-art results, the idea is nice), but there are key weaknesses (e.g., it describes incremental work), and it can significantly benefit from another round of revision. However, I won't object to accepting it if my co-reviewers champion it.

**Missing References:**

None.

**Paper Topic And Main Contributions:**

This paper focuses on domain adaptation tasks and aims to make model generalize better for unseen domains. The key idea is using hypernetworks to generate the classifier weights. They also provide two variations: with example-based domain related features (DRF) and with example-based DRF + prompt. Experimental results show the effectiveness of the proposed methods.

**Questions For The Authors:**

- What intrinsic mechanism in hypernetworks makes them useful?
- Does the architecture of hypernetworks affect the performance a lot?
- You mention that PADA is the current state-of-the-art model. But it performs not well in CLCD setting (Table 2). Any possible reasons?

**Reasons To Accept:**

- The improvements are promising.
- Considering hypernetworks is interesting.

**Reasons To Reject:**

- The motivation is not clear enough for me. Although the experimental results look good, the authors should answer the question that why hypernetworks help generalization. What intrinsic mechanism in hypernetworks makes them useful?
- It's unclear for me that why adding prompt help the generalization. I know it's proposed by the PADA paper, but I suggest the authors provide more insights or experiments to study more about it.

**Reproducibility:**

3: Could reproduce the results with some difficulty. The settings of parameters are underspecified or subjectively determined; the training/evaluation data are not widely available.

**Reviewer Confidence:**

3: Pretty sure, but there's a chance I missed something. Although I have a good feel for this area in general, I did not carefully check the paper's details, e.g., the math, experimental design, or novelty.

---

> ### Author Rebuttal · Authors · 2023-08-29
>
> **General response to all reviewers:**\
> We thank the reviewers for their insightful comments.
> As acknowledged by most reviewers, we address an important and challenging task of adaptation to unknown domains under a low resource regime. This task has been explored in the literature, and the best-performing approaches are PADA and MoE. Our proposed methods achieve strong results in extensive experimentation, outperforming PADA and MoE by impressive gaps.
> We would like to list again our main algorithmic contributions in short:
> 1. The use of Hypernetworks (HNs) for adaptation to unseen domains (i.e. domains that are unknown at training time).
> 2. The application of HNs in an example-based manner. Using our mechanism, the HNs’ output is tailored to each example, which allows a more flexible model and improves performance.
> 3. The integration of the prompting mechanism into a hyper-network-based framework, in a way that yields SOTA adaptation performance.
>
> **Response to reviewer suhr:**
>
> 1. The motivation for Hypernetworks:
> The effectiveness of hypernetworks in domain adaptation can be attributed to their ability to transcend discrete domain boundaries and embrace a continuous perspective. Unlike conventional methods that often focus solely on one domain, hypernetworks enable a more holistic approach, allowing each sample to draw upon the collective knowledge of all pertinent domains (an approach like MoE that does consider multiple domains, imposes strict domain boundaries).  In our implementation, the knowledge share is at the example level,  where each example is represented through a mixture of domain-related semantic components.
>
> 2. The motivation for Prompts:
> As demonstrated in extensive NLP research, prompts help put things in context for the model. In our case, the prompt aims to put the input example (from an unfamiliar domain) in a semantic context that the model recognizes (I.e. as noted above, the prompt represents the example as a mixture of domain-related semantic components).
>
> In practice, the prompts are used to condition the discriminative model on the specific domain attributes presented in them. For instance, when the prompt is made of the domain name (like in the Hyper-DN model), the classifier is conditioned on the domain and learns different P(Y|X;D) distributions for each domain D. When the prompt contains DFRs (like in Hyper-DRF and Hyper-PADA), each example is represented through  an example-specific mixture of source domain properties. This forces the classifier to learn a more complex conditional distribution and enables the classifier to generalize to examples from unseen domains.
>
> The PADA paper explores this issue extensively, providing evidence and analysis of the importance of the prompt. For sanity, we checked for similar behaviors in our methods (e.g., what happens when the prompt consists of random tokens), and observed similar trends to those reported in PADA. However, we do not think that these analyses should be presented in our paper, as we would like to save valuable space for novel material. Given the opportunity we will address this in the final version.
>
>
> 3. The effect of the hypernetwork architecture:
> Following this comment, we experimented with 3 different architectural configurations and measured the variation of scores.
> The architecture of the HN in the paper consisted of two linear layers with an additional activation on top of each. To estimate the effect of the architecture on the downstream results, we also test a 1-layer-based HN and a 3-layer-based HN. We report the results of this experiment in the following table, focusing on the MNLI task.
>
> MNLI (CD)\
> | | #HN Layers | #params| F | G | S | TL | TR | AVG |\
> +------------------+-----------+--------+-----+-----+-----+-----+-----+-----+\
> | Hyper-PADA | 1 | 2.4M |79.2 |83.7 |76.5 |80.2 |79.1 |79.7 |\
> | Hyper-PADA | 2 | 3.0M |79.0 |84.1 |78.2 |79.8 |81.1 |80.4 |\
> | Hyper-PADA | 3 | 3.5M |77.8 |83.4 |77.7 |79.5 |80.9 |79.9 |\
> +------------------+-----------+--------+-----+-----+-----+-----+-----+-----+
>
>
>
>  As demonstrated in the above table, the effect of the HN architecture exists, yet is not prominent, at least with the experimented changes. The robustness to hyper-parameter configurations is hence another upside of our method.
>
>
> 4. PADA’s weak performance in CLCD:
>
> Our method combines the advantages of both prompting and hypernetworks, while PADA utilizes only prompting. In the CLCD setup, which was not part of the PADA paper, it's evident that hypernetworks are more effective than prompting, based on the following observations:
> 1. Our hypernetwork (HN) models significantly outperform all models that do not employ an HN.
> 2. There's only a minor performance difference between our HN-based  models with varying prompting strategies.
> 3. As noted by the reviewer, PADA performs poorly on this task. We believe the diversity of languages disrupts PADA's prompting mechanism, thus affecting its efficiency.
>
>
> Notice that our CLCD setup - adaptation to unseen domains and languages - have not been explored in the past in the NLP literature (to the best of our knowledge). Hence, uncovering the reduced performance of PADA in this setup is a contribution of our paper. Yet, as PADA is so strong in the cross-domain setup, it was a straightforward decision to consider it for CLCD as well. We will make this clear in the final version.

---

### Official Review · Reviewer_4JUK · 2023-08-05

**Soundness:** 3

**Excitement:**

3: Ambivalent: It has merits (e.g., it reports state-of-the-art results, the idea is nice), but there are key weaknesses (e.g., it describes incremental work), and it can significantly benefit from another round of revision. However, I won't object to accepting it if my co-reviewers champion it.

**Missing References:**

[1] Zhao, Sicheng, et al. "Multi-source distilling domain adaptation." Proceedings of the AAAI Conference on Artificial Intelligence. Vol. 34. No. 07. 2020.

[2] Mansour, Yishay, Mehryar Mohri, and Afshin Rostamizadeh. "Domain adaptation with multiple sources." Advances in neural information processing systems 21 (2008).

[3] https://www.cs.jhu.edu/~mdredze/datasets/sentiment/

**Paper Topic And Main Contributions:**

The authors propose solutions for a multiple source domain adaptation scenario where the target domain is not accessible during the training. Their solutions are based on hyper-networks. In the first case the hyper-network learns a representation for each domain and generates the weights of the classifier. In the second case, the hyper-network learns a representation for each target example using a T5 model and generates the classifier weights. And in the third case the hyper-network learns the representation and a prompt.

They test their models in two tasks and 4 languages and show that it outperforms some naive baselines—the task is fairly new, so not many baselines are available.

**Reasons To Accept:**

The task is practical. Typographically, the paper is well-written and easy to understand. The models are straightforward, and the improvements are convincing. The experiments are also thorough.

**Reasons To Reject:**

The methods are very straightforward—in a good way—not much can be said about them, but some of the arguments in the paper are weak, vague, and incorrect. In Lines 89-101 the authors claim that “there is no DA method that focuses on domain invariant and domain specific features simultaneously.”
There has been more more than 20 years of research on domain adaptation, of course there has been studies on that!! See [1]. The reality is that because the task that the authors are working on is very new, they can claim that in that specific task this idea has not been used—but yet no other ideas has been used either, the task is new anyway.

In Lines 283-294, the authors use the argument that “because weight sharing may not be sufficient and because domain boundaries are not always well defined,” then they propose a model to learn from the target examples.
I don’t understand this argument. If you are arguing that a model or solution is not working well, you need to reveal its weakness. Show its flaws. Not just use vague words. Domain adaptation is a theoretical field. Indeed there have been theories that prove there exists a function consisting of the weighted average of the source distributions that has arbitrary low target error [2]. That function may be achieved by weight sharing.
If you have an idea that works, and you do not know why it works, please just say you don’t know why it works. Avoid giving vague arguments.

Please avoid forcing reviewer to read appendix, see Line 320.

Lines 408-417, I am not convinced by the authors argument about the choice of dataset. Please see the newer version of the famous amazon dataset for domain adaptation [3].



[1] Zhao, Sicheng, et al. "Multi-source distilling domain adaptation." Proceedings of the AAAI Conference on Artificial Intelligence. Vol. 34. No. 07. 2020.

[2] Mansour, Yishay, Mehryar Mohri, and Afshin Rostamizadeh. "Domain adaptation with multiple sources." Advances in neural information processing systems 21 (2008).

[3] https://www.cs.jhu.edu/~mdredze/datasets/sentiment/

**Reproducibility:**

4: Could mostly reproduce the results, but there may be some variation because of sample variance or minor variations in their interpretation of the protocol or method.

**Reviewer Confidence:**

3: Pretty sure, but there's a chance I missed something. Although I have a good feel for this area in general, I did not carefully check the paper's details, e.g., the math, experimental design, or novelty.

---

> ### Author Rebuttal · Authors · 2023-08-29
>
> **General response to all reviewers:**\
> We thank the reviewers for their insightful comments.
> As acknowledged by most reviewers, we address an important and challenging task of adaptation to unknown domains under a low resource regime. This task has been explored in the literature, and the best-performing approaches are PADA and MoE. Our proposed methods achieve strong results in extensive experimentation, outperforming PADA and MoE by impressive gaps.
> We would like to list again our main algorithmic contributions in short:
> 1. The use of Hypernetworks (HNs) for adaptation to unseen domains (i.e. domains that are unknown at training time).
> 2. The application of HNs in an example-based manner. Using our mechanism, the HNs’ output is tailored to each example, which allows a more flexible model and improves performance.
> 3. The integration of the prompting mechanism into a hyper-network-based framework, in a way that yields SOTA adaptation performance.
>
>
> **Response to reviewer 4JUK:**\
> We appreciate your recognition of our task's importance, the improvements our models present, and our work's thorough empirical evaluation. We believe that addressing your concerns will sharpen our message and improve our paper. Below, we address the concerns raised:
>
> 1. Vague arguments
> - Lines 89-101 - Thank you. Indeed we meant that the statement is correct in the current context. We will clarify the scope, and novelty, and refer to [1].
> - Lines 283-294 - We think our argument was not understood. Following this comment, we will replace the vague explanation in line  283 with: “We hypothesize that parameter sharing in the domain level may lead to suboptimal performance. Following [2], we present the Hyper-DRF model.
>
> 2. Choice of the dataset.
> Indeed, the famous Amazon dataset for domain adaptation is an example of a very large dataset with a diverse set of domains. Yet, this dataset is exceptional as it is tied to a major application of one oof the world's largest companies and the labels are generated by Amazon's users. Hence, we still think that our argument is correct for the much more common scenario: when addressing a new task, one typically has a limited annotation budget (caused by budget/time/data-availability). In such cases, they practically have to choose between a large number of examples from a single domain and a smaller number of examples from several domains. We will clarify that this is not always the case, and cite projects like the Amazon sentiment analysis dataset.
>
>
> [1] Zhao, Sicheng, et al. "Multi-source distilling domain adaptation." Proceedings of the AAAI Conference on Artificial Intelligence. Vol. 34. No. 07. 2020.
> [2] Ben-David, Eyal, et al. "PADA: Example-based Prompt Learning for on-the-fly Adaptation to Unseen Domains."

---

### Official Review · Reviewer_StMx · 2023-08-09

**Soundness:** 3

**Excitement:**

3: Ambivalent: It has merits (e.g., it reports state-of-the-art results, the idea is nice), but there are key weaknesses (e.g., it describes incremental work), and it can significantly benefit from another round of revision. However, I won't object to accepting it if my co-reviewers champion it.

**Paper Topic And Main Contributions:**

This paper proposes a new approach using hypernetworks for domain adaptation in natural language processing. The key contributions are:

1. Proposes three models (Hyper-DN, Hyper-DRF, Hyper-PADA) using hypernetworks for domain adaptation, allowing parameter sharing at the domain and example levels. This is a novel application of hypernetworks in NLP.
2. Introduces an example-based adaptation method where the model generates a "signature" for each example to map it to the semantic space of the source domains. This signature is used as input to the hypernetwork and to modify the input representation.
3. Evaluates the models extensively on sentiment analysis and natural language inference across multiple domains and languages. The results show the hypernetwork models outperform previous domain adaptation methods.


**Reasons To Accept:**

1. Novel application of hypernetworks for domain adaptation in NLP: The paper presents an innovative way to leverage hypernetworks for parameter sharing in domain adaptation scenarios. This opens up new possibilities for using hypernetworks in NLP.
2. Thorough empirical evaluation: The models are evaluated extensively across multiple domain adaptation setups over two tasks involving both cross-domain and cross-lingual transfer. This provides convincing evidence of the benefits of the proposed approach.

**Reasons To Reject:**

1. Limited to classification tasks: The experiments focus only on text classification tasks. It is unclear if the approach would transfer well to other NLP tasks like sequence labeling, generation, etc. Expanding the evaluation to other tasks could strengthen the paper.
2. Requires large pretrained models: The approach relies on large pretrained encoder-decoder models like T5, which have significant computational requirements. Especially for Hyper-DRF  and Hyper-PADA, they require an additional T5 model. Exploring ways to make the approach more efficient could broaden its applicability.
3. Incremental contribution: The core idea of using hypernetworks for domain adaptation is not entirely new, though the specific application to unseen target domains in NLP is novel. The incremental contribution over prior hypernetwork adaptation approaches should be clarified.

**Reproducibility:**

3: Could reproduce the results with some difficulty. The settings of parameters are underspecified or subjectively determined; the training/evaluation data are not widely available.

**Reviewer Confidence:**

3: Pretty sure, but there's a chance I missed something. Although I have a good feel for this area in general, I did not carefully check the paper's details, e.g., the math, experimental design, or novelty.

---

> ### Author Rebuttal · Authors · 2023-08-29
>
> **General response to all reviewers:**\
> We thank the reviewers for their insightful comments.
> As acknowledged by most reviewers, we address an important and challenging task of adaptation to unknown domains under a low resource regime. This task has been explored in the literature, and the best-performing approaches are PADA and MoE. Our proposed methods achieve strong results in extensive experimentation, outperforming PADA and MoE by impressive gaps.
> We would like to list again our main algorithmic contributions in short:
> 1. The use of Hypernetworks (HNs) for adaptation to unseen domains (i.e. domains that are unknown at training time).
> 2. The application of HNs in an example-based manner. Using our mechanism, the HNs’ output is tailored to each example, which allows a more flexible model and improves performance.
> 3. The integration of the prompting mechanism into a hyper-network-based framework, in a way that yields SOTA adaptation performance.
>
>
> **Response to reviewer StMx:**\
> We appreciate your recognition of our work's novelty and its thorough empirical evaluation. Below, we address the concerns raised. We note that the first two concerns raised by the reviewer were also discussed in our paper, under the Limitations section.
>
> 1. Limited to Classification Tasks:
>
> We stress that our setup choice, adapting to unfamiliar domains and languages, is already advanced. As our experiments demonstrate, many well-established domain adaptation methods do not perform well in our scenarios. Diving into other tasks, like token classification, would introduce more complexities. For instance, token-based classification would force discerning the nuances of applying hypernetworks at the token level and the heightened challenge of adapting to unseen domains for token-based tasks. While these challenges are undeniably pivotal, addressing them would demand dedicated research that might shift the focus of our current paper.
>
> Notice also that our work addresses two tasks, a large number of domains and both cross-domain and cross-language adaptation.  This is already a much more extensive experimental setup than in most domain adaptation work. In the PADA paper, for example, which has an unusally extensive experimental setup, there is a token classifcation task, but no cross-language generalization is demonstrated; Indeed in our paper we show that PADA does not perform well in the cross-language cross-domain adaptation setup, and our HN-based models perform much better.
>
>  Nonetheless, we acknowledge the significance of this comment and have earmarked it for future exploration.
>
> 2. Requires Large Pretrained Models:
>
> To put things in perspective, in the age of hundreds of billions parameter models (one of them, GPT3.5, is serving as a baseline to our models), our largest model employs two-scale fewer parameters. Furthermore, MoE, one of the previous SOTA models for our setup, requires more parameters than ours.
>
> Following this comment, we experimented with the following models, testing the effect of making our models more efficient:
> - Hyper-PADA with a single T5-large (instead of two T5-base).
> - Hyper-PADA with a single T5-base.
> - Hyper-DRF with a single T5-small.
> - Hyper-PADA with two T5-small architectures.
>
> Overall, large models improve end-to-end results, and using two models is superior to using one. However, our model is superior to all baselines even with a single T5-base. For convenience, we report the results achieved with Hyper-PADA variants and compare them to those documented in the paper:
>
> MNLI (CD)\
> |                 | #T5   | model size |  F  |  G  |  S  |  TL |  TR | AVG |\
> +------------------+-------+------------+-----+-----+-----+-----+-----+-----+\
> | Hyper-PADA      |   2   |   small    |67.1 |71.9 |61.6 |67.6 |67.0 |67.0 |\
> | Hyper-PADA      |   1   |   Small    |69.3 |75.6 |65.6 |69.1 |71.7 |70.3 |\
> +------------------+-------+------------+-----+-----+-----+-----+-----+-----+\
> | Hyper-PADA      |   2   |   Base     |79.0 |84.1 |78.2 |79.8 |81.1 |80.4 |\
> | Hyper-PADA      |   1   |   Base     |78.7 |84.1 |76.8 |78.1 |80.7 |79.7 |\
> +------------------+-------+------------+-----+-----+-----+-----+-----+-----+\
> | Hyper-PADA      |   2   |   Large    |85.4 |88.1 |83.7 |85.8 |88.1 |86.2 |\
> | Hyper-PADA      |   1   |   Large    |85.6 |88.1 |79.8 |85.9 |86.8 |85.2 |
>
> 3. Incremental Contribution:
> Previous works in the intersection of HNs and DA in NLP assume limited labeled data in the target domain. Using this data, with much-labeled data in other domains, the HN learns to generate the set of parameters for the target domain. Notice that this setup is arguably not a domain adaptation challenge in the context of emerging few-shot learning models. Hence, this contribution is very limited compared to ours.
>
> Our contribution over prior HN approaches is three-fold:
> -  Our method is tailored for scenarios without prior target domain knowledge. This presents unique challenges, especially in determining the appropriate weights the hypernetwork should generate. Notice that the number of papers that addressed our domain adaptation scenario is limited and our results are substantially stronger.
> - We integrate into our model a 'signature' generation mechanism for each example (borrowed from the PADA paper), and this signature is fed both to the representation componenet and to the HN. This enables a novel application of an HN-based model in an example-based manner  - something that has not been done in the past in the field of NLP. The resulting strong performance are much better than those of the original PADA algorithm.
> - We are the first to present an example-based hypernetwork approach (that is, even regardless of our contribution to out-of-distribution generalization), particularly in NLP.
>
>  We will make sure to highlight these contributions in the final version, given the opportunity.

---

### Official Review · Reviewer_cySR · 2023-08-10

**Soundness:** 2

**Excitement:**

3: Ambivalent: It has merits (e.g., it reports state-of-the-art results, the idea is nice), but there are key weaknesses (e.g., it describes incremental work), and it can significantly benefit from another round of revision. However, I won't object to accepting it if my co-reviewers champion it.

**Paper Topic And Main Contributions:**

This paper addresses the issue of multi-source adaptation for unfamiliar domains: We leverage labeled data from multiple source domains to generalize to unknown target domains at training. Authors employ example-based Hypernetwork adaptation: a T5 encoder-decoder initially generates a unique signature from an input example, embedding it within the source domains' semantic space.

**Questions For The Authors:**

1. Authors state "hypernetworks in example-based manner, which is novel at least in NLP, to the best of our knowledge". So, is it

**Reasons To Accept:**

1. The paper is well-written.

2. The idea seems to be interesting.

3. The paper studies an important problem.

**Reasons To Reject:**

1. The proposed method is a little naive, which combines T5 encoder-decoder with a hypernetwork.

2. There could be some different methods to learn shared and domain-specific aspects of the input such as disentanglement. Why the proposed is better?

3. More visualization and interpretability and  should be provided to validate the effectiveness of the hypernetwork.

**Reproducibility:**

4: Could mostly reproduce the results, but there may be some variation because of sample variance or minor variations in their interpretation of the protocol or method.

**Reviewer Confidence:**

2: Willing to defend my evaluation, but it is fairly likely that I missed some details, didn't understand some central points, or can't be sure about the novelty of the work.

---

> ### Author Rebuttal · Authors · 2023-08-29
>
> **General response to all reviewers:**\
> We thank the reviewers for their insightful comments.
> As acknowledged by most reviewers, we address an important and challenging task of adaptation to unknown domains under a low resource regime. This task has been explored in the literature, and the best-performing approaches are PADA and MoE. Our proposed methods achieve strong results in extensive experimentation, outperforming PADA and MoE by impressive gaps.
> We would like to list again our main algorithmic contributions in short:
> 1. The use of Hypernetworks (HNs) for adaptation to unseen domains (i.e. domains that are unknown at training time).
> 2. The application of HNs in an example-based manner. Using our mechanism, the HNs’ output is tailored to each example, which allows a more flexible model and improves performance.
> 3. The integration of the prompting mechanism into a hyper-network-based framework, in a way that yields SOTA adaptation performance.
>
> **Response to reviewer cySR:**\
> We appreciate the positive feedback regarding the quality of writing, the novelty of the idea, and the significance of the problem addressed. We have carefully considered your concerns and provide the following comments:
>
> 1. Regarding the naive method:
>
> Recapping our method: Hyper-PADA uses a T5 encoder-decoder to generate a unique (example specific) semantic signature. Then, this signature is fed as a prefix of the example into the T5 encoder, and as an input to the HN which generates the classifier weights for this specific example. Finally, the classifier predicts the task label conditioned on the encoding vectors of the ‘prompted’ input.
>
> The development of our method required nuanced intuition and a deep understanding of the problem, informed by extensive familiarity with prior works. Moreover, the uniqueness of our approach in this context underlines its innovation.
>
> Finally, and perhaps most importantly, our results are very strong, across tasks, domains and languages, which is indicative of the quality of our model. As our ablation analysis shows, many alternatives to our model perform much worse.
>
> 2. Alternative methods like disentanglement:
> Disentanglement approaches may serve as an efficient way to learn domain-specific and domain-invariant properties of the data. Following this comment, we adjusted the implementation of [1] to fit our setup. Specifically, we learn domain-generic and domain-specific spaces and use both to predict the task label. The domain generic space is learned with a domain adversarial loss (instead of NSP, as in [1], to fit the domain scenario) and the domain-specific space is learned in a similar fashion as in [1], through domain classifier (instead of task classifier). We tested this approach on our MNLI snd Sentiment CLCD (comparing to our models):
>
> MNLI (CD)\
> | |  F    |  G    |  S    |  TL  |  TR | AVG |\
> +-------------------+-----+-----+-----+-----+-----+-----+\
> | Disentangle |69.5 |77.1 |67.8  |71.9  |72.8 |71.8 |\
> | Hyper-PADA |79.0 |84.1  |78.2  |79.8  |81.1  |80.4 |\
> +-------------------+-----+-----+-----+-----+-----+-----+
>
>
> Sentiment (CLCD)\
> |                |Deutsch             |English             |French              |Japanese           | AVG |\
> |                |  B  |  D  |  M  |  B  |  D  |  M  |  B  |  D  |  M  |  B  |  D  |  M  |      |\
> +----------------+------+------+------+------+------+------+------+------+------+------+------+------+\
> | Disentangle |82.7 |80.0 |82.6 |87.3 |82.7 |83.4 |80.2 |53.8 |80.2 |68.2 |82.9 |58.2 |76.8 |\
> | Hyper-PADA |85.7 |81.8 |85.0 |86.0 |84.4 |85.1 |86.8 |85.9 |81.8 |83.9 |83.9 |83.8 |84.5 |\
> +----------------+------+------+------+------+------+------+------+------+------+------+------+------+
>
>
> As can be seen, our models are superior to this approach.
>
>
> 3. Visualization & interpretability:
> We kindly refer the reviewer to the experiments presented in Appendix C. Specifically, in lines 1262-1283 (Figure 6), we reveal the significance of diversity in the weights generated by the HN, highlighting the mechanism that  allows Hyper-PADA to outperform PADA.
> Following this comment, we will take two steps:\
> (A) We will move into the extra page of main paper (allowed in the camera ready version) the following analyses currently in the appendix.
> - The importance of diversity in generated weights (Fig 6)
> - The effect of training size on performance (Fig 5)
> - Performance metrics for seen domains (Table 6)
>
> (B) To shed light on the success of our method for different HN architectures, we add an experiment considering 1/2/3 layers HNs. Our results:
>
> MNLI (CD)\
> |                 | #HN Layers | #params|  F  |  G  |  S  |  TL |  TR | AVG |\
> +------------------+-----------+--------+-----+-----+-----+-----+-----+-----+\
> | Hyper-PADA      |     1     | 2.4M   |79.2 |83.7 |76.5 |80.2 |79.1 |79.7 |\
> | Hyper-PADA      |     2     | 3.0M   |79.0 |84.1 |78.2 |79.8 |81.1 |80.4 |\
> | Hyper-PADA      |     3     | 3.5M   |77.8 |83.4 |77.7 |79.5 |80.9 |79.9 |
>
>
> ** Questions:
> Yes, we are the first to present an example-based hypernetwork in NLP.
>
> [1] “Continual Learning for Text Classification with Information Disentanglement Based Regularization”, Huang et al., 2021.

---

### Meta-Review · Area_Chair_A3P7 · 2023-09-18

**Recommendation:** 4

**Metareview:**

All the reviewers agree that this method paper is well-written, works on a practical problem, and presents a convincing improvement for unseen domain adaptation given the extensive experimental setup (across domains, tasks, and languages). One common weakness of the paper are the motivation for the usage of hypernetworks, which manifests in different ways such as "Why not consider disentanglement" and "Why hypernetworks help generalization?"  In my opinion, I think that the authors have done a good job in the rebuttals in motivating the use of hypernetworks, explaining the advantages of their method, and showing the results comparing their proposed method and the disentanglement method mentioned by reviewer cySR. The performance gap between their proposed method and the existing methods such as PADA and MoE is quite impressive.

While the reviewers raise other issues on the simplicity of the approach and the lack of other types of tasks and datasets, in my opinion, this paper has already demonstrated a promising approach to tackle a non-trivial task that they set out to explore and carried out extensive experiments including ablation study to verify the strength of their proposed approach. While it is undesirable that many findings are in the Appendix, such as the ablation and interpretability experiments, this paper is information-heavy and yet the authors have done a good job in articulating the settings and their important results. However, certain arguments in the paper could have been strengthened as suggested by reviewer 4JUK. The motivation for the proposed approach should also be detailed further in the revised version of the paper.

---

### Decision · Program_Chairs · 2023-10-07

**Decision:**

Accept-Findings

**Comment:**

All the reviewers agree that this method paper is well-written, works on a practical problem, and presents a convincing improvement for unseen domain adaptation given the extensive experimental setup (across domains, tasks, and languages). One common weakness of the paper are the motivation for the usage of hypernetworks, which manifests in different ways such as "Why not consider disentanglement" and "Why hypernetworks help generalization?"  In my opinion, I think that the authors have done a good job in the rebuttals in motivating the use of hypernetworks, explaining the advantages of their method, and showing the results comparing their proposed method and the disentanglement method mentioned by reviewer cySR. The performance gap between their proposed method and the existing methods such as PADA and MoE is quite impressive.

While the reviewers raise other issues on the simplicity of the approach and the lack of other types of tasks and datasets, in my opinion, this paper has already demonstrated a promising approach to tackle a non-trivial task that they set out to explore and carried out extensive experiments including ablation study to verify the strength of their proposed approach. While it is undesirable that many findings are in the Appendix, such as the ablation and interpretability experiments, this paper is information-heavy and yet the authors have done a good job in articulating the settings and their important results. However, certain arguments in the paper could have been strengthened as suggested by reviewer 4JUK. The motivation for the proposed approach should also be detailed further in the revised version of the paper.